# A Framework for Standardizing Similarity Measures in a Rapidly Evolving Field

**Nathan Cloos**
MIT
nacloos@mit.edu

**Guangyu Robert Yang**
MIT
yanggr@mit.edu

**Christopher J. Cueva**
MIT
ccueva@gmail.com

## Abstract

Similarity measures are fundamental tools for quantifying the alignment between artificial and biological systems. However, the diversity of similarity measures and their varied naming and implementation conventions makes it challenging to compare across studies. To facilitate comparisons and make explicit the implementation choices underlying a given code package, we have created and are continuing to develop a Python repository that benchmarks and standardizes similarity measures. The goal of creating a consistent naming convention that uniquely and efficiently specifies a similarity measure is not trivial as, for example, even commonly used methods like Centered Kernel Alignment (CKA) have at least 12 different variations, and this number will likely continue to grow as the field evolves. For this reason, we do not advocate for a fixed, definitive naming convention. The landscape of similarity measures and best practices will continue to change and so we see our current repository, which incorporates approximately 100 different similarity measures from 14 packages, as providing a useful tool at this snapshot in time. To accommodate the evolution of the field we present a framework for developing, validating, and refining naming conventions with the goal of uniquely and efficiently specifying similarity measures, ultimately making it easier for the community to make comparisons across studies.

**Code:** https://github.com/nacloos/similarity-repository

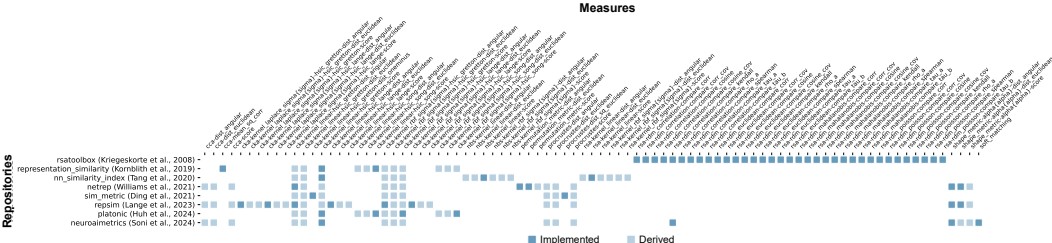

Figure 1: **We register and standardize existing implementations of similarity measures to facilitate comparisons across studies.** Dark blue entries are the similarity measures originally implemented in the GitHub repositories associated with their respective papers: [Kriegeskorte et al., 2008a], [Kornblith et al., 2019], [Tang et al., 2020], [Williams et al., 2021], [Ding et al., 2021], [Lange et al., 2022], [Huh et al., 2024], [Soni et al., 2024]. We aim to include all papers that provide code for similarity measures but here we show only a subset of the packages implemented in the full online repository. Light blue entries indicate additional measures that we can automatically derive from the original ones, thereby expanding the grounds for comparison.

38th Conference on Neural Information Processing Systems (NeurIPS 2024).

# 1 Introduction

Similarity measures have become a cornerstone in evaluating representational alignment across different models [Kornblith et al., 2019], different biological systems [Kriegeskorte et al., 2008b], and across both artificial and biological systems. Researchers have employed diverse methods to compare model representations with, for example, brain activity, aiming to identify models that exhibit brain-like representations [Yamins et al., 2014, Sussillo et al., 2015, Schrimpf et al., 2018, Nayebi et al., 2018]. However, while these measures are actively used and provide an efficient way to compare structure across complex systems, it is not clear that they adequately represent the computational properties of interest, and there is a need to better understand their limitations [Cloos et al., 2024]. The field lacks clear guidelines for selecting the most appropriate measure for a given context.

As the number of proposed similarity measures continues to grow, the landscape becomes more complex and fragmented. Reviews like Sucholutsky et al. [2023], Klabunde et al. [2023] attempt to summarize and categorize these measures, but they often do not address the full computational diversity or nuances of each method, which is not always apparent unless the code implementation is reviewed in detail.

To facilitate comparisons and make explicit the implementation choices underlying a given code package, we have created and are continuing to develop a Python repository that standardizes similarity measures. Our aims for the repository are:

1. **Register and standardize existing implementations:** We compile implementations from various papers and map them to a standardized naming convention that aims for **consistency**, **low naming complexity**, and **reflects the underlying mathematical structure**. Importantly, a naming convention that maps out the current space of similarity measures will not continue to be definitive as these measures and best practices continue to evolve. So our goal is not to advocate for a static naming convention but to provide a repository that usefully specifies current similarity measures, while also providing a framework for continuing to refine these naming conventions as the field evolves.

2. **Serve as a centralized reference:** The repository provides a centralized place where practitioners can easily find which methods each paper implemented and exactly how they compare across studies via quantitative comparisons.

3. **Facilitate development of new implementations:** By providing accessible code references, we make it easier for practitioners to write their own implementations and validate them through quantitative comparisons with existing ones.

We demonstrate the utility of our approach using Centered Kernel Alignment (CKA) as an illustrative example.

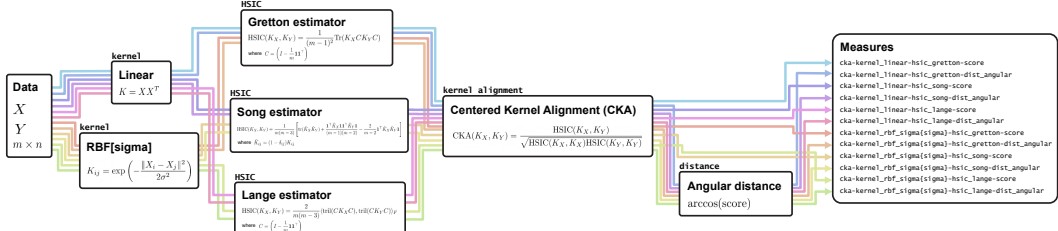

Figure 2: **Our standardized naming convention unifies existing variations of CKA measures.** The convention is designed to reflect the underlying compositional structure by highlighting the kernel method, the HSIC estimator, and the scoring method as the three main components that constitute the various CKA measures. Standardized names separate these components with a dash symbol. See Appendix A.2 for details.

# 2 Methods

**Standardization procedure**. To develop our package we iterate through the following steps:

1. **Collecting implementations:** Identify all similarity measures implemented in the GitHub repositories associated with relevant papers.

2. **Understanding interfaces:** Analyze the input and output interfaces for each implemented function and convert inputs to the expected format (e.g., transposing or centering data arrays if necessary).

3. **Mapping to standardized names:** Map each implemented function to a standardized name that reflects its mathematical components, starting with simple names and adding complexity as needed (see step 4).

4. **Validating consistency:** Test the consistency of outputs across implementations assigned the same name. If inconsistencies are found, refine the naming convention to capture necessary distinctions.

## 3   Results

**Standardizing CKA implementations**. As an illustrative example, we applied our standardization procedure to implementations of CKA from 5 different repositories associated with [Kornblith et al., 2019], [Tang et al., 2020], [Williams et al., 2021], [Lange et al., 2022], [Huh et al., 2024]. We found 12 different names were necessary to capture the distinct variations of CKA (see Figure 2 and Appendix A.2 for more details).

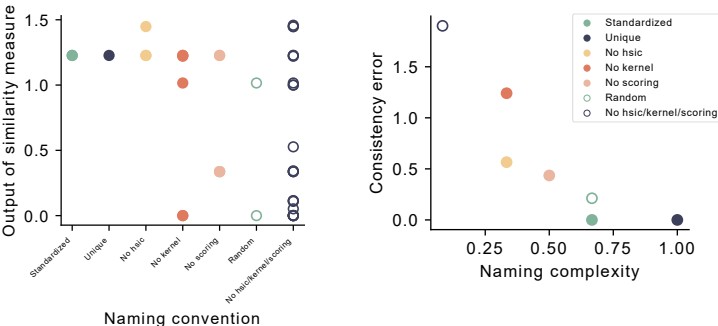

Figure 3: **Our standardized naming convention achieves both low complexity and consistency error of zero.** (Left) The output of the similarity measure can vary widely if the name is not fully specified. All scores shown here are computed between the same two datasets and the variability derives solely from variations in CKA. (Right) Summary of the consistency error across all datasets and measure names for each naming convention. Naming conventions with fewer names (lower complexity) have higher error due to assigning the same name to measures that produce different scores (see Appendix A.4 for details).

**Implementation choices can have large impacts.** Choices for CKA measures are not just subtleties of the implementation but can have a large impact on the final similarity score. For example, Figure 3 (left) shows the variability across CKA scores when two datasets are compared using different variations of CKA. All the variability is due to variations of CKA.

**Achieving both low complexity and consistency error of zero.** Figure 3 (left) demonstrates the variability in similarity values when evaluating different naming conventions on a single pair of datasets (see Appendix A.3 for details on the datasets). Out of the naming conventions considered, only our standardized convention and the "unique" naming convention - which assigns a unique name to each implementation - produce consistent similarity values for all measures sharing the same name.

Figure 3 (right) summarizes the consistency error across all datasets and measure names for each naming convention. Naming conventions with fewer names than the standardized one (lower complexity), such as when removing specifiers from the standardized names ("No hsic", "No kernel", "No scoring", "No hsic/kernel/scoring" in Figure 3), result in higher consistency error. This indicates that these conventions assign the same name to different measures that produce varying scores. The "unique" naming convention has a consistency error of zero but has a higher naming complexity than the standardized one (see Appendix A.4 for details).

**Papers implement a small and distinct subset of the CKA measures.** After standardizing the implementations from various studies, mapping them to a common set of names and validating their consistency, we visualize and compare the specific measures they implement (Figure 4).

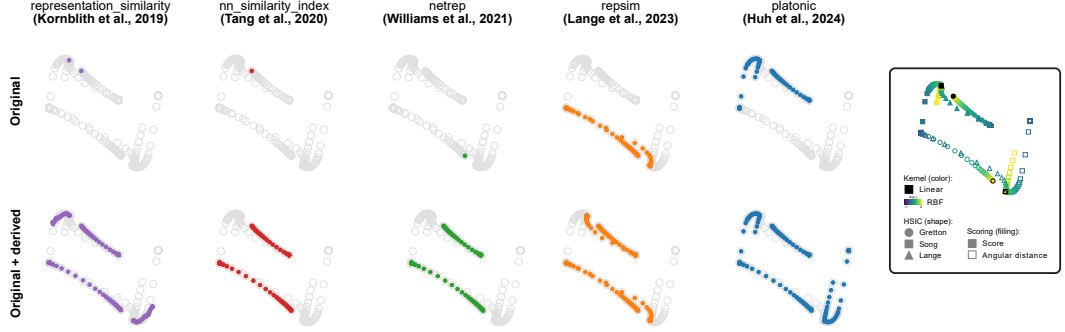

Figure 4: **Our standardized repository allows users to compare similarity scores across studies and derive new variations that were not initially implemented.** (Top row) Each colored dot shows the variations of CKA that were implemented in the listed studies, and summarizes the relationships when comparisons are made across multiple datasets (Supplementary Figure 9) visualized in 2D using multidimensional scaling. Many studies, and corresponding code packages, only implement a distinct subset of the similarity measures, making it challenging to compare results across studies. (Bottom row) Colored dots show the originally implemented CKA variations plus the variations that can be derived. Not all variations are derivable from each individual study (see Appendix A.6 for details).

**Are all measures equivalent?** Do papers really need to implement all the variants of CKA? Are the different measures just on different scales and need to have the right conversion, for example, a linear scaling to transform between them? No. See Figures 4 and 5.

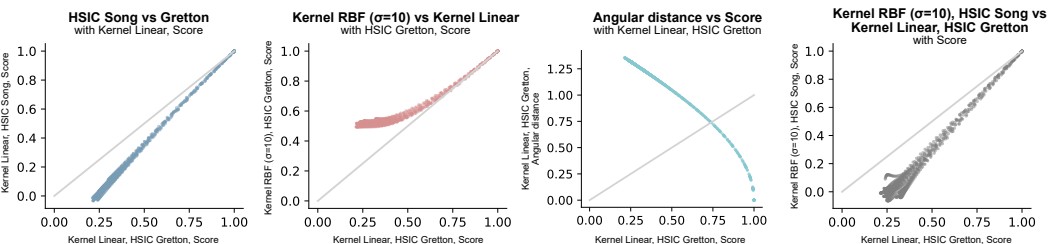

Figure 5: **Are variations of CKA capturing the same information? No.** Each plot shows comparisons between two variations of CKA across multiple datasets. The lack of consistency suggests the different variations provide different windows into the similarity between datasets. Gray line shows y=x for comparison.

## 4 Discussion

Comparing similarity scores across studies is challenging, primarily due to variability in naming and implementation conventions. As part of our contribution to the research community we have created, and are continuing to develop, a Python repository that benchmarks and standardizes similarity measures. The ideal naming convention for similarity measures should flexibly evolve to incorporate new similarity measures and adapt as we change our best practices. To facilitate this evolution, we outline our framework for developing, validating, and refining naming conventions to standardize implementations. We hope our work is a step towards building tools for more reproducible and integrative science.

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

# A  Appendix

## A.1  Python repository

Our Python repository simplifies the use and comparison of similarity measures implemented in various research papers. To compare different measures from the same package users can easily call `similarity.make` with the package name and the similarity measure's name as illustrated in Figure 6. A similar call to `similarity.make` also allows users to compare different implementations of the same measure. By standardizing the naming of similarity measures across different packages using the procedure proposed in this work, we facilitate comparisons across studies.

Furthermore, users can register their own implementations of existing or new similarity measures, as shown in Figure 7. Once registered, these custom measures can be used just like any other measure in the package. Registering your own measures enables you to compare your implementations with existing ones, which is particularly useful for validating new implementations and ensuring they produce consistent results.

```python
import numpy as np
import similarity

# different measures from the same package
measure1 = similarity.make("netrep/procrustes-dist_angular")
measure2 = similarity.make("netrep/cka-kernel_linear-hsic_gretton-dist_angular")

X = np.random.randn(40, 20)
Y = np.random.randn(40, 20)

score1 = measure1(X, Y)
score2 = measure2(X, Y)

# same measure but different packages
implementation1 = similarity.make("netrep/procrustes-dist_angular")
implementation2 = similarity.make("repsim/procrustes-dist_angular")

score1 = implementation1(X, Y)
score2 = implementation2(X, Y)
```

Figure 6: Our repository makes it easy to use multiple measures from a given package or the same measure from different packages, facilitating comparisons across different studies.

## A.2  Family of CKA similarity measures

We demonstrate the utility of our approach using Centered Kernel Alignment (CKA) as an illustrative example. CKA is a similarity measure that quantifies the alignment between two matrices $X$ and $Y$ by measuring the Hilbert-Schmidt Independence Criterion (HSIC) between two kernel matrices $K_X$ and $K_Y$ [Kornblith et al., 2019].

$$\text{CKA}(K_X, K_Y) = \frac{\text{HSIC}(K_X, K_Y)}{\sqrt{\text{HSIC}(K_X, K_X)\text{HSIC}(K_Y, K_Y)}}$$

This alignment score ranges between 0 and 1, where 1 corresponds to perfect similarity.

A common choice of kernel is the linear kernel, given by:

$$K_X = XX^\top \quad \text{and} \quad K_Y = YY^\top$$

assuming that the columns of $X$ and $Y$ are centered (i.e., have zero mean). Here, $X$ and $Y$ are matrices of shape (samples, features). CKA with linear kernel is invariant to orthogonal transformations applied along the feature dimension, which is desirable when aligning representations that may differ by rotation.

```python
import numpy as np
import similarity

# register your own measure
@similarity.register("my_package/cka-kernel_linear-hsic_gretton-score")
def my_measure(X, Y):
    # your code
    return 0

# or without the decorator
similarity.register("my_package/cka-kernel_linear-hsic_gretton-score", my_measure)

# then use it as any other measure
X = np.random.randn(40, 20)
Y = np.random.randn(40, 20)

my_measure = similarity.make("my_package/cka-kernel_linear-hsic_gretton-score")
my_score = my_measure(X, Y)
print("My score:", my_score)

# and compare it to existing implementations
measures = similarity.make("*/cka-kernel_linear-hsic_gretton-score")
for name, measure in measure.items():
    score = measure(X, Y)
    print(f"{name}: {score}")
```

Figure 7: Users can easily register their own implementations and compare them with implementations of the same measure from different papers.

Another common kernel choice is the radial basis function (RBF) kernel, defined as:

$$K_{ij} = \exp\left(-\frac{\|X_i - X_j\|^2}{2\sigma^2}\right)$$

where $\sigma$ controls the importance of large distances over small distances.

HSIC is a measure of statistical dependence between two random variables, often estimated with the empirical estimator [Gretton et al., 2005]:

$$\text{HSIC}(K_X, K_Y) = \frac{1}{(m-1)^2} \text{Tr}(K_X C K_Y C)$$

where $m$ is the number of samples, and $C = I - \frac{1}{m}\mathbf{1}\mathbf{1}^\top$ is the centering matrix ($I$ is the identity matrix and $\mathbf{1}$ is a vector of ones).

Song et al. [2007] introduced an unbiased estimator of HSIC, defined as:

$$\text{HSIC}(K_X, K_Y) = \frac{1}{m(m-3)}\left[\text{Tr}(\tilde{K}_X \tilde{K}_Y) + \frac{\mathbf{1}^\top \tilde{K}_X \mathbf{1} \mathbf{1}^\top \tilde{K}_Y \mathbf{1}}{(m-1)(m-2)} - \frac{2}{m-2}\mathbf{1}^\top \tilde{K}_X \tilde{K}_Y \mathbf{1}\right]$$

where $\tilde{K}_{ij} = (1 - \delta_{ij})K_{ij}$ is the kernel matrix with diagonal entries set to zero.

More recently, Lange et al. [2022] proposed an estimator with lower bias, of order $O(m^{-2})$ compared to the $O(m^{-1})$ bias of the Gretton estimator:

$$\text{HSIC}(K_X, K_Y) = \frac{2}{m(m-3)} \langle \text{tril}(C K_X C), \text{tril}(C K_Y C)\rangle_F$$

where $\text{tril}(A)$ denotes the vector formed by the elements of the lower triangular part of matrix $A$, excluding the diagonal, and $\langle \cdot, \cdot \rangle_F$ denotes the Frobenius inner product.

Furthermore, Williams et al. [2021] proposed taking the arccosine of CKA to obtain a measure satisfying the axioms of a distance metric, referred to as Angular CKA by Lange et al. [2022].

## A.3   Synthetic datasets

We quantitatively compare the different variations of CKA on synthetic datasets to evaluate the consistency of the various naming conventions. Specifically, we randomly sample a Gaussian dataset

$X$ with an exponentially decaying eigenspectrum with values ranging from 10 to 0.1. The dataset $X$ has shape $(40, 20)$, representing 40 examples with 20 features each.

From this dataset $X$, we create a sequence of datasets $Y$ by progressively adding standard Gaussian noise to $X$. We considered 10 different noise levels with standard deviations ranging from 0 to 10. This results in datasets $(X, Y)$ with varying degrees of similarity: maximal similarity when $Y = X$, decreasing as more noise is introduced. We repeat this process for 10 independently generated datasets $X$, producing a total of 100 different dataset pairs.

## A.4 Comparison of naming conventions

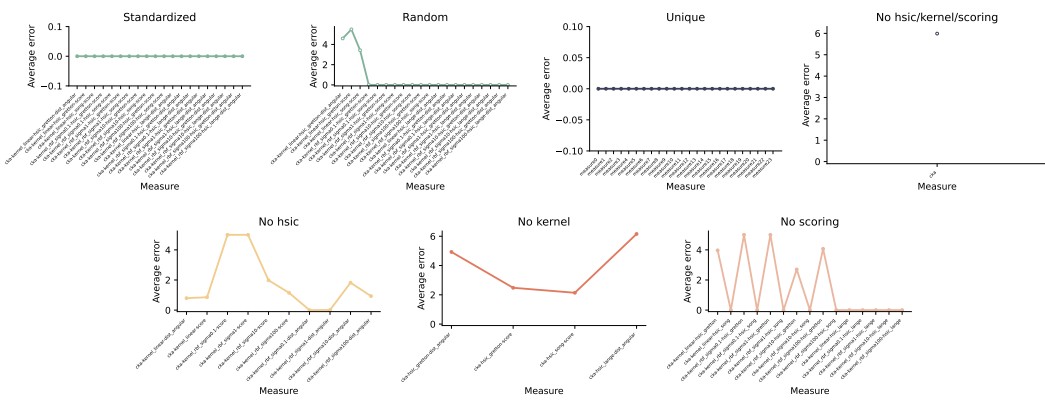

Figure 8: Average pairwise mean squared error (MSE) between similarity measures assigned the same name under different naming conventions.

For a given set of code packages and respective implementations of similarity measures, we define a naming convention as a mapping that assigns names to implementations.

We evaluate naming conventions along two axes:

- **Naming complexity:** Ratio of the number of distinct names divided by the total number of implementations across the different packages. A naming convention that assigns the same name to all the implementations has the lowest complexity. The maximal value for the complexity is one and is achieved when each implementation is assigned a unique name.
- **Consistency error:** The average error between the scores produced by measures assigned the same name when evaluated on the same set of datasets. We estimate this quantity by measuring the mean squared error (MSE) and using the synthetic datasets described in Appendix A.3.

The optimal naming convention should have both low naming complexity to efficiently encode variations of similarity measures, and consistency error of zero to ensure implementations assigned the same name produces identical results. Taking CKA as an example, the naming complexity for the optimal naming convention is the ratio of the number of distinct names required to uniquely specify the variations of CKA divided by the total number of implementations of CKA across the different packages.

We evaluate several naming conventions for similarity measures to assess the trade-off between naming complexity and consistency error:

- **Standardized**: Uses the full standardized naming convention including all specifiers.
- **No hsic**: Removes the HSIC estimator specifier.
- **No kernel**: Removes the kernel specifier.
- **No scoring**: Removes the scoring method specifier.
- **No hsic/kernel/scoring**: Removes all specifiers, which amounts to using a single name (i.e., "CKA") for all measures.

- **Random**: Randomly shuffles names from the standardized convention.
- **Unique**: Assigns a unique name to each implementation.

Figure 8 presents the average pairwise MSE between the outputs of similarity measures sharing the same name across the different naming conventions. The labels on the x-axis represent the different names within each naming convention. Ideally, similarity measures with the same name should yield identical outputs when applied to the same inputs, corresponding to a zero mean squared error.

Naming conventions with fewer names (lower complexity), such as "No hsic," "No kernel," "No scoring," or "No hsic/kernel/scoring," result in higher consistency error. This is because these conventions assign the same name to different measures that produce varying scores. The "Random" naming convention also exhibits high error due to mismatched naming.

The results reported in Figure 3 (right plot) represent the average of the values plotted in Figure 8 for each naming convention, where each dot in Figure 3 (right plot) is the average of one plot in the supplementary figure.

## A.5    Comparison of CKA measures

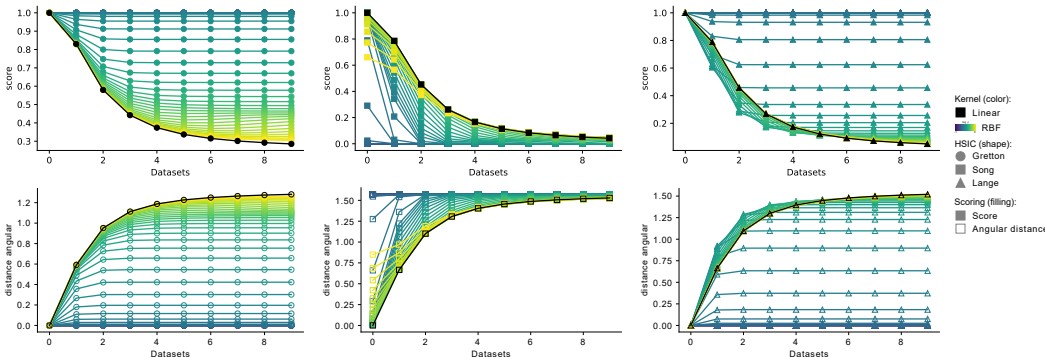

Figure 9: Different variations of CKA give different outputs when evaluated on the same datasets. CKA measures vary along three main components, the kernel method, the HSIC estimator, and the scoring method. We compare the different combinations of these components by evaluating them on 10 pairs of synthetic datasets. The synthetic datasets are constructed by randomly sampling a Gaussian dataset $X$ with an exponentially decaying egeinspectrum, and creating a sequence of datasets $Y$ by adding standard Gaussian noise to $X$ with increasing variance (see Appendix A.3) Each curve corresponds to a specific variant of CKA, which corresponds to a point in the MDS plot of Figure 4. The dissimilarity matrix for MDS is computed from the pairwise mean squared error between these curves.

## A.6    Deriving additional similarity measures from original implementations

In some instances, we can easily convert one variant of a similarity measure into another by applying transformations to the inputs or outputs of the implemented measures. For example, applying the arccosine function to the output of the score version of CKA gives the angular distance version of CKA. However, this method is not always applicable as it relies on the specific mathematical properties of the measures. For example, to our knowledge, there is not straightforward way to derive the unbiased HSIC estimator proposed by Song et al. [2007] from the biased HSIC proposed by Gretton et al. [2005] solely by modifying the inputs or outputs. When applicable, this approach can conveniently extend the original set of implemented measures in a given repository but may not be enough to cover all the possible variants.

The transformations we have implemented include:

- Applying the arccosine function to derive CKA with angular distance from CKA score, and vice versa with the cosine function.

- Deriving CKA measures with an RBF kernel from an implementation with a linear kernel by first computing the RBF kernels of the inputs $(X, Y)$, then taking the matrix square root so that computing the linear kernel as $XX^\top$ recovers the RBF kernel of $X$, and similarly for $Y$.

- Converting between Euclidean and angular distances. Given the angular distance $\theta$ between $X$ and $Y$, the Euclidean distance $d$ can be computed as:

$$d = \sqrt{\|X\|_F^2 + \|Y\|_F^2 - 2\|X\|_F\|Y\|_F \cos(\theta)}$$

The angular distance $\theta$ can be computed from $d$ as:

$$\theta = \arccos\left(\frac{\|X\|_F^2 + \|Y\|_F^2 - d^2}{2\|X\|_F\|Y\|_F}\right)$$

- Identifying measures that are the same but have different names. For example, shape metrics with $\alpha = 0$ corresponds to Canonical Correlation Analysis (CCA) and $\alpha = 1$ corresponds to orthogonal Procrustes.

