# OpenReview forum: "A Framework for Standardizing Similarity Measures in a Rapidly Evolving Field"
_NeurIPS.cc/2024/Workshop/UniReps — UniReps_

### Official Review · Reviewer_sq3B · 2024-10-02

**Rating:** 8
**Confidence:** 2

**Review:**

This work develops a common framework for implementing various similarity metrics. Using Centered Kernel Alignment (CKA) as their main example, they first demonstrate that there can be large variability in similarity measures depending on the specific choice of CKA variant used. Further, they show that different CKA variants are not equivalent suggesting they capture different measures of similarity between representations.

Strengths: This paper addresses a challenge in the area of representational alignment: there are a growing number of representational similarity measures, often implemented in distinct repositories. A single well-maintained repository that standardizes implementation is an important resource for the community.

Relevance: This paper is clearly relevant for this workshop, I recommend acceptance.

Comments:
- I found Figure 4 very difficult to parse. Are these methods applying different version of CKA on different datasets? In that case, I don't understand how this comparison is being implemented.
- While different similarity metric can capture different information about systems, there are formal relationships between these measures (e.g., equivalences, bounds) that are being established in the literature and it may be useful to note these equivalences as the package is being developed (see Harvey et al. "Duality of Bures and Shape Distances with Implications for Comparing Neural Representations" 2023).

---

### Official Review · Reviewer_bPko · 2024-10-04
**A valiant effort to unify and standardize comparisons of neural representational geometry**

**Rating:** 8
**Confidence:** 5

**Review:**

The authors present a Python package that implements a large number of network representational similarity measures that have been independently proposed. Even more importantly, they develop a useful naming convention and framework for classifying different distance measures.

I think this is a great idea and will be a useful conversation starter at the workshop. The main NeurIPS track does not do a good job of supporting work like this that synthesizes the literature and provides straightforward user implementations of common methods.

I suggest that the authors add a citation to a related benchmark that appeared recently: https://arxiv.org/abs/2408.00531

---

### Official Review · Reviewer_mvd1 · 2024-10-05
**This paper makes a substantial contribution to the field by proposing a standardized framework for similarity measures. It is original, practically useful, and significant in its potential to improve reproducibility and efficiency in research. With minor adjustments to enhance clarity and accessibility, it could have an even broader impact.**

**Rating:** 6
**Confidence:** 4

**Review:**

Quality

The paper tackles a critical issue in comparing artificial and biological systems through similarity measures. The proposed Python repository for standardizing these measures is a practical and valuable tool. The methodology is rigorous, and the examples, particularly with Centered Kernel Alignment (CKA), are well-illustrated.

Clarity

While the paper is generally clear and well-organized, sections dealing with technical details (e.g., naming conventions) may overwhelm readers unfamiliar with similarity measures. Simplifying some parts could improve accessibility.

Originality

The paper is innovative in its approach to solving the growing complexity of similarity measures across studies. Creating an evolving naming convention is a novel and useful idea, adding significant value to research reproducibility.

Significance

The work's significance lies in its potential to standardize comparisons across different studies, enhancing reproducibility and efficiency in machine learning and neuroscience research. The framework’s adaptability ensures its long-term relevance.

Pros
1. Relevant and timely contribution to the field.
2. Practical tool for researchers through the Python repository.
3. Flexible and adaptable framework for future developments.

Cons
1. High complexity in technical details may limit accessibility.
2. More examples across different domains could strengthen the paper.

---

### Official Review · Reviewer_mYYR · 2024-10-07
**Introducing a new repository to unify the implementation of similarity measures, providing an opportunity for comprehensive comparisons.**

**Rating:** 6
**Confidence:** 4

**Review:**

Summary :
The authors introduce a new repository which compiles various implementations of the similarity measures from different sources, and support the importance of their work by empirical results.


Strengths :
*  Authors do not introduce a new method, but compile existing implementations into a unified repository, addressing potential incompatibilities across various research work. Relevant previous work is thoroughly cited.
* The claims about similarity measure incompatibilities across various works are supported by experiments. Although it is a work in progress, it holds promise to help researchers with implementing the appropriate similarity measure and making the process more efficient.
* Research statement is clearly explained, and the results are reproducible given the details regarding the dataset generation in the Appendix A.3, the computation of the metrics in Appendix A.4 and the link to the GitHub repository.
* Results indicate the importance of the unified choice of similarity measures among various studies to ease the comparison and benchmarking.
* Anonymity is preserved with the GitHub link and contributions to the repository.



Weaknesses / Suggestions:
* The repository could be improved by including theoretical explanations and guidance to help researchers select the appropriate similarity measure.
* The authors do not introduce any novel implementations, but rather compile various existing implementations under one platform.
* The importance and potential issues caused by different implementations of the same similarity measure are not thoroughly explained, which somewhat diminishes the overall significance of the work.
* The paper does not sufficiently discuss the limitations of the work.

---

### Decision · Program_Chairs · 2024-10-10

**Decision:**

Accept (Oral)

**Comment:**

In light of the positive reviewers' feedback and relevancy of the submission, we are pleased to accept this paper for presentation at UniReps 2024. We kindly ask the authors to incorporate the reviewers' suggestions and feedback in the final camera-ready version of the manuscript.